# Octopamine drives honeybee thermogenesis

Sinan Kaya-Zeeb[1], Lorenz Engelmayer[1], Mara Straßburger[1], Jasmin Bayer[2], Heike Bähre[3], Roland Seifert[3], Oliver Scherf-Clavel[2], Markus Thamm[1]*

[1]Behavioral Physiology and Sociobiology, Julius Maximilian University of Würzburg, Würzburg, Germany; [2]Institute for Pharmacy and Food Chemistry, Julius Maximilian University of Würzburg, Würzburg, Germany; [3]Institute of Pharmacology, Research Core Unit Metabolomics, Hannover Medical School, Hannover, Germany

**Abstract** In times of environmental change species have two options to survive: they either relocate to a new habitat or they adapt to the altered environment. Adaptation requires physiological plasticity and provides a selection benefit. In this regard, the Western honeybee (*Apis mellifera*) protrudes with its thermoregulatory capabilities, which enables a nearly worldwide distribution. Especially in the cold, shivering thermogenesis enables foraging as well as proper brood development and thus survival. In this study, we present octopamine signaling as a neurochemical prerequisite for honeybee thermogenesis: we were able to induce hypothermia by depleting octopamine in the flight muscles. Additionally, we could restore the ability to increase body temperature by administering octopamine. Thus, we conclude that octopamine signaling in the flight muscles is necessary for thermogenesis. Moreover, we show that these effects are mediated by β octopamine receptors. The significance of our results is highlighted by the fact the respective receptor genes underlie enormous selective pressure due to adaptation to cold climates. Finally, octopamine signaling in the service of thermogenesis might be a key strategy to survive in a changing environment.

## Editor's evaluation

This study is of broad interest to researchers in the field of entomology and physiology. These findings may shed light on at least one mechanism underlying selective advantages conferred to insect species on evolutionary timescales. Though the chemical signal, its source, and recipient tissues underlying thermogenesis are elucidated, hypotheses regarding their downstream effects remain to be substantiated.

*For correspondence: markus.thamm@uni-wuerzburg.de

**Competing interest:** The authors declare that no competing interests exist.

## Introduction

The Western honeybee (*Apis mellifera*) owns incredible thermoregulation strategies, which allow the colony to keep the brood area constantly at 34 °C (*Simpson, 1961*). Due to this special feature, honeybees are relatively independent of the ambient temperature ($T_A$), which may contribute decisively to their almost worldwide distribution (*Wallberg et al., 2014*). In contrast to other ectotherms, honeybee thermoregulation includes thermogenesis. Here, primarily workerbees actively increase their thorax temperatures ($T_{THX}$, *Kovac et al., 2009*; *Stabentheiner et al., 2010*). This thermogenesis is of immense social importance, because it enables foraging at $T_A$ below 10 °C (*Bujok et al., 2002*; *Stabentheiner et al., 2003*) and a proper brood development (*Himmer, 1932*; *Weiss, 1962*; *Tautz et al., 2003*; *Wang et al., 2016*), reduces parasite infections (*Starks et al., 2000*; *Campbell et al., 2010*), and is a powerful defense mechanism against predatory hornets (*Ken et al., 2005*; *Baracchi et al., 2010*).

The individual heating pattern of workerbees consists of a wave-like rise and fall in $T_{THX}$ (*Kronenberg and Heller, 1982*) and is realized exclusively by the activation of the indirect flight muscles, formed by the dorsoventral wing elevators (DV) and the dorsal-longitudinal wing depressors (DL), even if wing and thorax vibration are not visible (*Esch et al., 1991*; *Esch and Goller, 1991*). However, these muscles are utilized in various other behaviors, which includes flight (*Esch et al., 1975*; *Esch, 1976*), fanning (*Simpson, 1961*) and communication during the waggle dance (*Esch, 1961*; *Wenner, 1962*). In order to perform these various tasks, diverse contraction mechanisms exist which must be controlled differently (*Esch and Goller, 1991*). Some evidence indicates a crucial role of octopamine in the insect flight muscles (*Blau and Wegener, 1994*; *Blau et al., 1994*; *Wegener, 1996*; *Duch et al., 1999*). Unfortunately, it remains unknown whether octopamine is used as a neurochemical in honeybee flight muscles or whether an octopamine receptor gene is expressed in these tissues. However, DL and DV are under control of the mesometa-thoracic ganglion (MMTG, *Markl, 1966*) and the octopaminergic innervation of the flight muscles seems to be a conserved feature in insects (*Duch et al., 1999*; *Schlurmann and Hausen, 2003*; *Pauls et al., 2018*). It was further demonstrated that the brain octopamine concentration of workerbees is significantly decreased due to cold stress (*Chen et al., 2008*), which indicates the temperature sensitivity of the neuronal octopaminergic system. In this context, *Wallberg et al., 2017* made the observation that honeybee β octopamine receptor genes (*AmOARβ1-3/4*) are subject to altitudinal adaptation processes in honeybees. Yet, the physiological significance of this result has not been investigated so far. One important parameter that decreases significantly with increasing altitude is $T_A$. Consequently, honeybee thermogenesis is essential for colony survival, and the adaptive pressure on *AmOARβ1-3/4* may indicate the involvement of octopamine in this process.

We hypothesize that honeybee thermogenesis relies on octopamine signaling and that β octopamine receptors are crucially involved in this process. We have investigated systematically the honeybee thoracic octopaminergic system. Moreover, we have tested our hypothesis and we can show that octopamine promotes thermogenesis by directly affecting the flight muscles.

## Results

### Honeybee flight muscles are innervated by octopaminergic neurons

First of all, we investigated whether octopamine can be a potential regulator of flight muscle functions in honeybees. Thus, we analyzed which monoamines are actually present in these tissues using

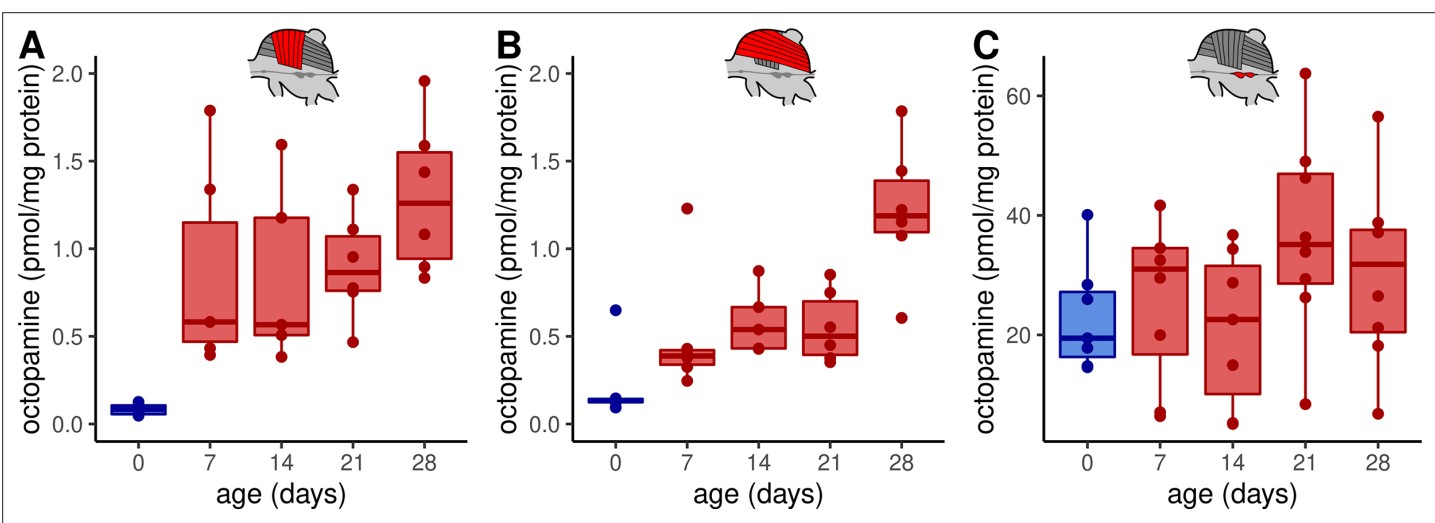

**Figure 1.** Octopamine concentrations in thoracic tissues across age. Octopamine concentrations differ significantly between different workerbee age groups in DV (**A**) and DL (**B**) but not in the MMTG (**C**). blue = no active heating, red = active heating. Shown is median ± interquartile range (IQR). For statistics see **Table 1**.

The online version of this article includes the following figure supplement(s) for figure 1:

**Figure supplement 1.** Monoamine quantification in workerbee thoracic tissues.

high-performance liquid chromatography (HPLC) together with an electrochemical detector (ECD). We can detect octopamine and dopamine in both, DV and DL, whereas serotonin and tyramine are not detectable (*Figure 1A*, *Figure 1—figure supplement 1A*). We further compared the flight muscle octopamine concentration in differently aged workerbees. Newly emerged bees which cannot perform thermogenesis have the lowest octopamine concentration in DV and DL (*Figure 1A–B*) and the octopamine concentration increases with the age of the workerbee (*Figure 1A–B*). In contrast to octopamine, the concentrations of dopamine have a different time course in DV and DL (*Figure 1—figure supplement 1B-C*). We have further analyzed the MMTG. In addition to octopamine, serotonin, dopamine, and tyramine are also detectable, but no age-related differences can be observed for any of these monoamines (*Figure 1C*, *Figure 1—figure supplement 1D-F*).

Nerves originating from the MMTG exclusively innervate the honeybee flight muscles (*Markl, 1966*; *Pan, 1980*). To answer whether octopamine in DV and DL can be delivered directly by octopaminergic neurons from the MMTG we used an octopamine specific antibody to analyze the octopamine distribution in these tissues. Octopamine-like immunoreactivity (OA-IR) is observable in four individual cell clusters, with most of the cell bodies being found at the ventral midline (*Figure 2A–E*). Some OA-IR positive cell bodies are also located at the dorsal midline (*Figure 2D–E*). Most MMTG leaving nerves show OA-IR (*Figure 2G–I*), as varicose fibers in IIN1 and a thicker axonal bundle in IIN3 demonstrate (*Figure 2G1*). Finally, finest OA-IR positive varicose structures can be found directly at muscle fibers (*Figure 2J–K*).

## *AmOARβ2* is expressed in the flight muscles

We next determined which octopamine receptor genes are expressed in the workerbee flight muscle. The honeybee genome harbors five different genes that code for octopamine receptors and two additional genes encoding tyramine receptors. The respective receptor proteins are functionally characterized (*Blenau et al., 2000*; *Grohmann et al., 2003*; *Balfanz et al., 2014*; *Reim et al., 2017*; *Blenau et al., 2020*). We observe strong signals for PCR products for *AmOARα1* and *AmOARβ2*, weak DNA bands for *AmOARβ1* and *AmOARβ3/4*, and no amplification product in the case of *AmOARα2* and both tyramine receptor genes (*AmTAR1* & *AmTAR2*; *Figure 3A*). In addition, PCR products indicate the expression for all known honeybee octopamine and tyramine receptor genes in neural tissues (brain, MMTG).

We further determined the relative gene expression of the most promising candidates by quantitative Real Time PCR (qPCR, *Figure 3B–E*). *AmOARα1* and *AmOARβ2* expression can be observed in DV and DL in all age groups of workerbees. Here, relative expression increases with age, as shown by significant differences between newly emerged bees (0 days) and the three oldest groups.

## Octopamine is mandatory for honeybee thermogenesis

To investigate the consequences of octopamine missing in the flight muscles, we fed workerbees with reserpine. This drug has the ability to deplete vesicles on monoaminergic synapses (*Plummer et al., 1954*; *Cheung and Parmar, 2020*). The octopamine concentrations in DV and DL are significantly decreased due to our treatment (*Figure 4A–B*). In contrast, the dopamine concentration in the flight muscle seems not to be affected (*Figure 4—figure supplement 1*). The same is true for the concentrations of octopamine (*Figure 4C*) and of the other monoamines in the MMTG (*Figure 4—figure supplement 1*).

The reserpine feeding additionally causes hypothermia in both, nurse bees and forager bees (*Figure 4D*, *Table 3*). A preliminary screen with serotonin, dopamine, octopamine and tyramine revealed, that octopamine may reverse the reserpine effect (*Figure 4—figure supplement 3*). We were able to show that this octopamine effect is robust. We reversed the reserpine-induced hypothermia by injecting octopamine directly into the flight muscles (*Figure 4*, *Table 3*).

As stated above, we hypothesize that β octopamine receptors are crucially involved in honeybee thermogenesis. Via Gα$_s$ proteins, these receptors are positively coupled to membrane-bound adenylyl cyclases (mAC), which leads to an increase of the intracellular adenosine 3',5'-cyclic mono-phosphate (cAMP) concentration upon receptor activation (*Balfanz et al., 2014*). To control our hypothesis, we have repeated the reserpine experiment reported above. The reserpine induced hypothermia as well as the octopamine reversion of this effect are again clearly observable (*Figure 4E*, *Table 3*). We stopped thermography after 5 min and the bees were immediately flash-frozen to subsequently

**Table 1.** Statistical analysis of HPLC analysis of the octopamine content.

ns = not significant.

| Analysis | Test | Groups (n) | Result | |
|---|---|---|---|---|
| | Kruskal-Wallis test | | $\chi^2$ = 15.772, df = 4, p = 0,0033 | ** |
| | | 0 days (6) vs. 7 days (6) | Z = −2.4593, $p_{adj}$ = 0.1392 | ns |
| | | 0 days (6) vs. 14 days (6) | Z = −2.8856, $p_{adj}$ = 0.0391 | * |
| | | 0 days (6) vs. 21 days (6) | Z = −2.7217, $p_{adj}$ = 0.065 | ns |
| HPLC octopamine DV *Figure 1A* | | 0 days (6) vs. 28 days (6) | Z = −3.7382, $p_{adj}$ = 0.0017 | ** |
| | Dunns test | 7 days (6) vs. 14 days (6) | Z = 0.4263, $p_{adj}$ = 1.0 | ns |
| | | 7 days (6) vs. 21 days (6) | Z = 0.2623, $p_{adj}$ = 1.0 | ns |
| | | 7 days (6) vs. 28 days (6) | Z = 1.2789, $p_{adj}$ = 1.0 | ns |
| | | 14 days (6) vs. 21 days (6) | Z = 0.164, $p_{adj}$ = 1.0 | ns |
| | | 14 days (6) vs. 28 days (6) | Z = −0.8526, $p_{adj}$ = 1.0 | ns |
| | | 21 days (6) vs. 28 days (6) | Z = −1.0165, $p_{adj}$ = 1.0 | ns |
| | Kruskal-Wallis test | | $\chi^2$ = 16.292, df = 4, p = 0.0027 | ** |
| | | 0 days (6) vs. 7 days (6) | Z = −1.3117, $p_{adj}$ = 1.0 | ns |
| | | 0 days (6) vs. 14 days (6) | Z = −2.6561, $p_{adj}$ = 0.0791 | * |
| | | 0 days (6) vs. 21 days (6) | Z = −1.9019, $p_{adj}$ = 0.5718 | ns |
| HPLC octopamine DL *Figure 1B* | | 0 days (6) vs. 28 days (6) | Z = −3.8038, $p_{adj}$ = 0.0014 | ** |
| | Dunns test | 7 days (6) vs. 14 days (6) | Z = 1.3444, $p_{adj}$ = 1.0 | ns |
| | | 7 days (6) vs. 21 days (6) | Z = 0.5902, $p_{adj}$ = 1.0 | ns |
| | | 7 days (6) vs. 28 days (6) | Z = 2.4921, $p_{adj}$ = 0.127 | ns |
| | | 14 days (6) vs. 21 days (6) | Z = 0.7542, $p_{adj}$ = 1.0 | ns |
| | | 14 days (6) vs. 28 days (6) | Z = −1.1477, $p_{adj}$ = 1.0 | ns |
| | | 21 days (6) vs. 28 days (6) | Z = −1.9019, $p_{adj}$ = 0.5718 | ns |
| HPLC octopamine MMTG *Figure 1C* | Kruskal-Wallis test | | $\chi^2$ = 5.4912, df = 4, p = 0.2405 | ns |
| | groups (n): 0 days (7), 7 days (8), 14 days (7), 21 days (8), 28 days (8) | | | |

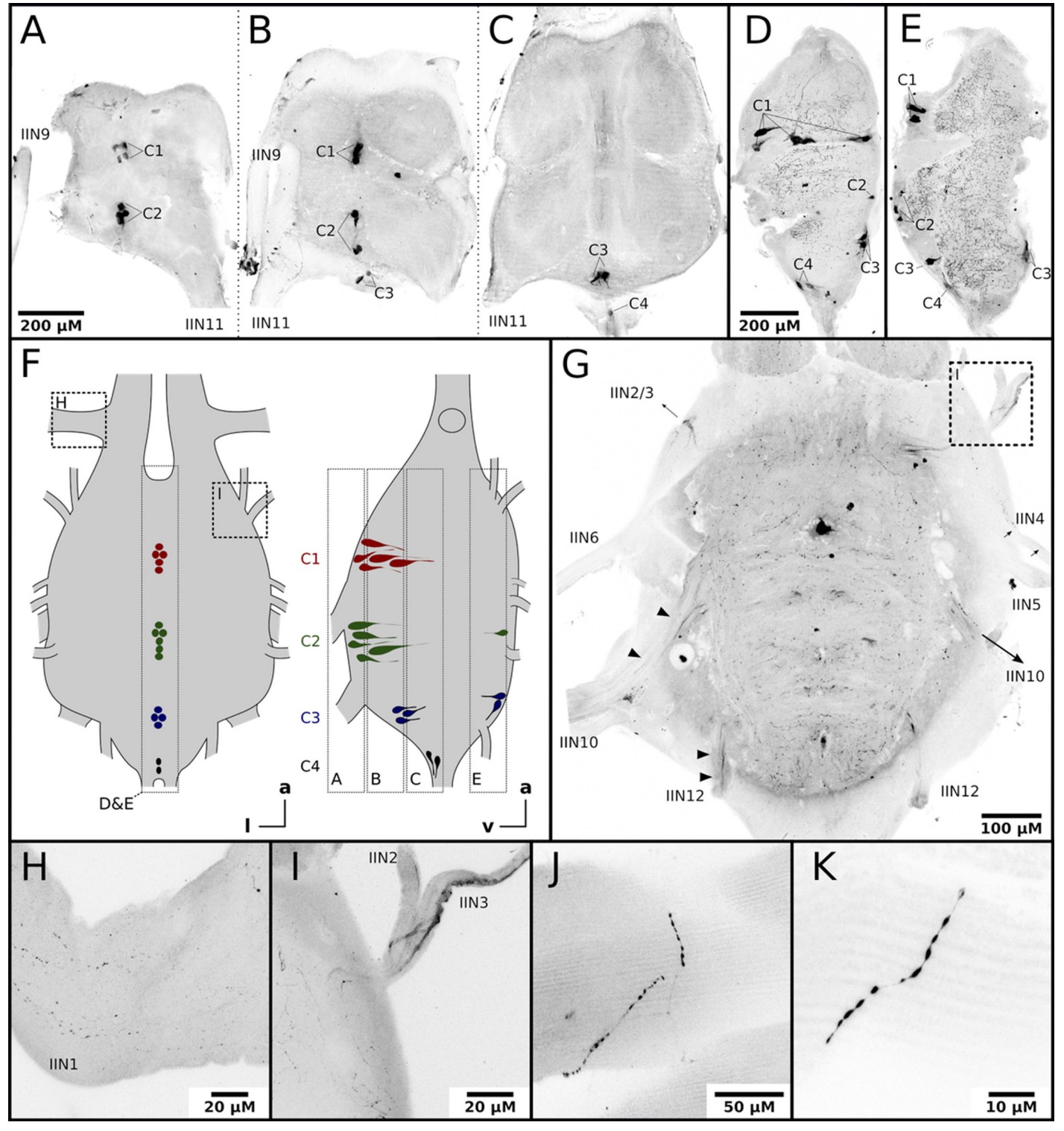

**Figure 2.** Honeybee flight muscles are innervated by octopaminergic neurons. (**A–E**) Different cell clusters with OA-IR are observable. Consecutive frontal sections of the MMTG of the same workerbee (**A–C**) beginning with the most ventral section (**A**) showing clusters of OA-IR positive cells (**C1–C4**). Sagital sections (**D–E**) in the midline area of the MMTG of two individual bees display the same OA-IR positive cell clusters. (**F**) Schematic interpretation of the location of the cell clusters found in A-E. Additionally, the approximate location of frontal sections (**A–C, G**), the sagital sections (**D–E**), and the detailed images (**H–I**) are indicated by dashed boxes. (**G**) Dorsally located frontal section of the MMTG in showing several nerves which are leaving the ganglion. Strong OA-IR-positive fibers run into the nerves IIN3, IIN10, and IIN12 (arrowheads). (**H**) Within the nerve IIN1 fine varicose

*Figure 2 continued on next page*

*Figure 2 continued*

structures with OA-IR are observable. (**I**) An OA-IR-positive axon bundle runs through the nerve IIN3. (**J–K**) Flight muscle preparations reveal fine varicose structures with OA-IR closely attached to muscle fibers.

quantify the tissue cAMP concentrations of their flight muscles. The tissue cAMP concentration is significant lower in reserpinized bees when compared with control (**Figure 4D**). Furthermore, octopamine injection into the flight muscles of reserpinized bees leads to a strong increase of the tissue cAMP concentration (**Figure 4D**). The tissue guanosine 3',5'-cyclic monophosphate (cGMP) concentrations seem not to be affected by our treatment (**Figure 4—figure supplement 4**). Further cyclic nucleotides in the flight muscles were either below the lower limit of quantification (cytidine 3',5'-cyclic monophosphate, cCMP) or were not detectable at all.

## Octopamine receptor antagonists also induce hypothermia

Next, we aimed to confirm the described effects of octopamine on honeybee thermogenesis and also to further narrow down the responsible receptor subtypes. Therefore, we injected different pharmacological substances directly into the flight muscles and analyzed their effect on thermogenesis. These substances antagonize various octopamine, tyramine, or adrenergic receptors. All antagonists either lead to hypothermia in both, nurse bees and forager bees, or they are not effective at all. The

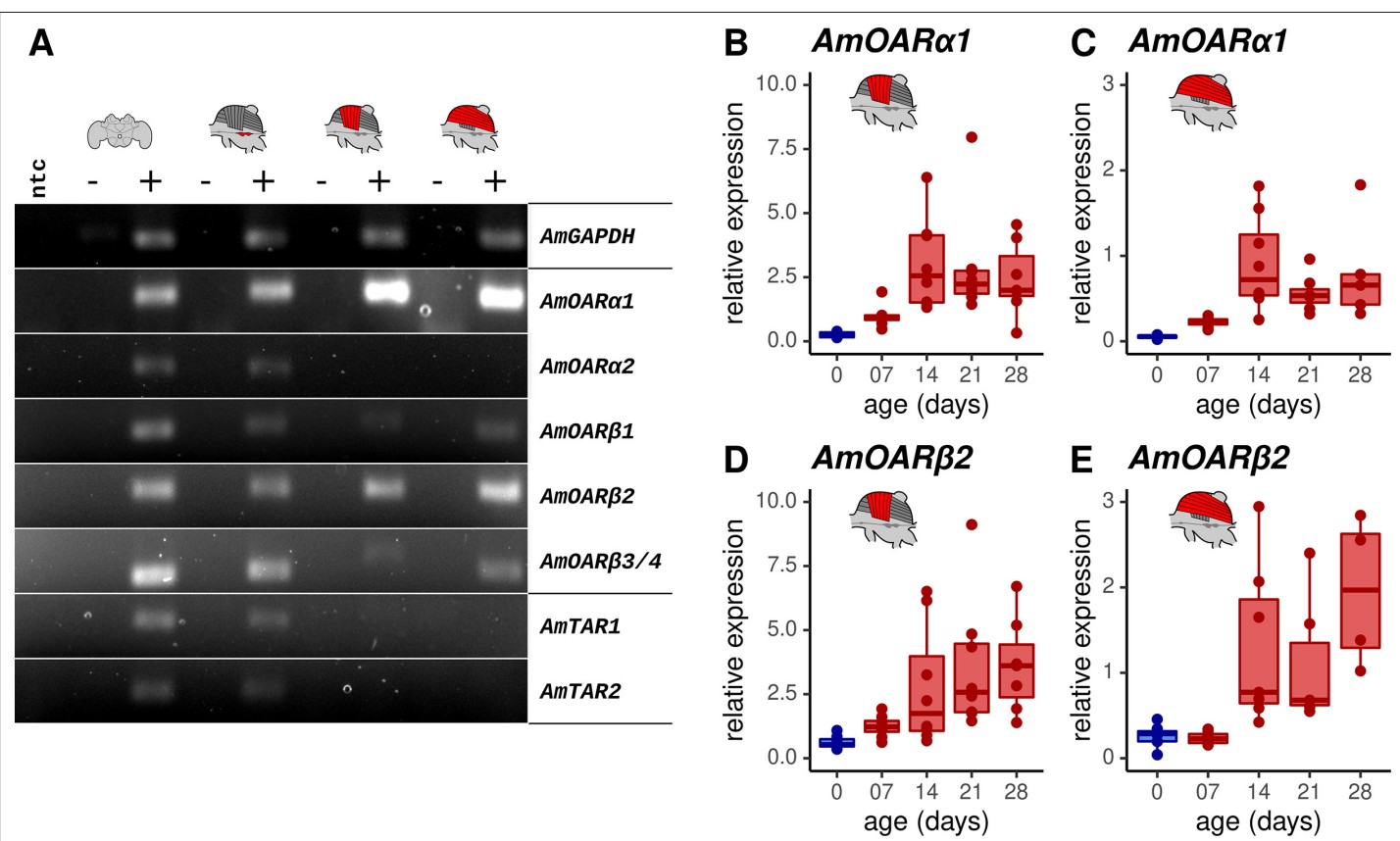

**Figure 3.** Octopamine receptor expression in the flight muscles. (**A**) Brain, MMTG, DV, and DL were manually dissected from workerbees and underwent subsequent RNA isolation, cDNA synthesis and PCR analysis (+). The reverse transcriptase was omitted during cDNA synthesis for negative controls (-). RNase free water serves as no template (ntc) and *AmGAPDH* as loading control. (**B–E**) *AmOARα1* and *AmOARβ2* expression in DV and DL of differential aged workerbees with (red) or without (blue) the capability for thermogenesis. Data are represented as boxplots. Shown is median ± IQR. For statistic see **Table 2**.

The online version of this article includes the following source data for figure 3:

**Source data 1.** Labelled original files of the full raw unedited PCR gels.

**Source data 2.** Unlabelled original files of the full raw unedited PCR gels.

**Table 2.** Statistical analysis of the flight muscle gene expression analysis.

ns = not significant.

| Analysis | Test | Groups (n) | Result | |
|---|---|---|---|---|
| qPCR AmOARα1 DV *Figure 3D* | Kruskal-Wallis test | | $\chi^2$ = 25.734, df = 4, p = 0.00004 | *** |
| | Dunns test | 0 days (8) vs. 7 days (8) | Z = –1.6253, $p_{adj}$ = 1.0 | ns |
| | | 0 days (8) vs. 14 days (8) | Z = –3.9776, $p_{adj}$ = 0.0007 | *** |
| | | 0 days (8) vs. 21 days (8) | Z = –3.9135, $p_{adj}$ = 0.0009 | *** |
| | | 0 days (8) vs. 28 days (8) | Z = –3.8493, $p_{adj}$ = 0.0012 | ** |
| | | 7 days (8) vs. 14 days (8) | Z = –2.3523, $p_{adj}$ = 0.1866 | ns |
| | | 7 days (8) vs. 21 days (8) | Z = –2.2882, $p_{adj}$ = 0.2213 | ns |
| | | 7 days (8) vs. 28 days (8) | Z = –2.224, $p_{adj}$ = 0.2615 | ns |
| | | 14 days (8) vs. 21 days (8) | Z = 0.0642, $p_{adj}$ = 1.0 | ns |
| | | 14 days (8) vs. 28 days (8) | Z = 0.1283, $p_{adj}$ = 1.0 | ns |
| | | 21 days (8) vs. 28 days (8) | Z = 0.0642, $p_{adj}$ = 1.0 | ns |
| qPCR AmOARα1 DL *Figure 3C* | Kruskal-Wallis test | | $\chi^2$ = 28.163, df = 4, p = 0.00001 | *** |
| | Dunns test | 0 days (8) vs. 7 days (8) | Z = –1.5661, $p_{adj}$ = 1.0 | ns |
| | | 0 days (8) vs. 14 days (8) | Z = –4.4373, $p_{adj}$ = 0.0001 | *** |
| | | 0 days (8) vs. 21 days (7) | Z = –3.6548, $p_{adj}$ = 0.0026 | ** |
| | | 0 days (8) vs. 28 days (5) | Z = –3.7128, $p_{adj}$ = 0.002 | ** |
| | | 7 days (8) vs. 14 days (8) | Z = –2.8712, $p_{adj}$ = 0.0409 | * |
| | | 7 days (8) vs. 21 days (7) | Z = –2.1418, $p_{adj}$ = 0.322 | ns |
| | | 7 days (8) vs. 28 days (5) | Z = –2.3392, $p_{adj}$ = 0193 | ns |
| | | 14 days (8) vs. 21 days (7) | Z = 0.6320, $p_{adj}$ = 1.0 | ns |
| | | 14 days (8) vs. 28 days (5) | Z = 0.179, $p_{adj}$ = 1.0 | ns |
| | | 21 days (7) vs. 28 days (5) | Z = –0.3844, $p_{adj}$ = 1.0 | ns |
| qPCR AmOARβ2 DV *Figure 3D* | Kruskal-Wallis test | | $\chi^2$ = 24.54, df = 4, p = 0.00006 | *** |
| | Dunns test | 0 days (8) vs. 7 days (8) | Z = –1.6894, $p_{adj}$ = 0,911 | ns |
| | | 0 days (8) vs. 14 days (8) | Z = –2.8228, $p_{adj}$ = 0.0476 | * |
| | | 0 days (8) vs. 21 days (8) | Z = –3.8707, $p_{adj}$ = 0.0011 | ** |
| | | 0 days (8) vs. 28 days (8) | Z = –4.3412, $p_{adj}$ = 0.0001 | *** |
| | | 7 days (8) vs. 14 days (8) | Z = –1.1334, $p_{adj}$ = 1.0 | ns |
| | | 7 days (8) vs. 21 days (8) | Z = –2.1813, $p_{adj}$ = 0.292 | ns |
| | | 7 days (8) vs. 28 days (8) | Z = –2.6517, $p_{adj}$ = 0.0801 | ns |
| | | 14 days (8) vs. 21 days (8) | Z = –1.0479, $p_{adj}$ = 1.0 | ns |
| | | 14 days (8) vs. 28 days (8) | Z = –1.5183, $p_{adj}$ = 1.0 | ns |
| | | 21 days (8) vs. 28 days (8) | Z = –0.4705, $p_{adj}$ = 1.0 | ns |

*Table 2 continued on next page*

*Table 2 continued*

| Analysis | Test | Groups (n) | Result | |
|---|---|---|---|---|
| | Kruskal-Wallis test | | $\chi^2$ = 24.737, df = 4, p = 0.00006 | *** |
| | | 0 days (8) vs. 7 days (8) | Z = 0.5429, $p_{adj}$ = 1.0 | ns |
| | | 0 days (8) vs. 14 days (7) | Z = −2.9652, $p_{adj}$ = 0.0302 | * |
| | | 0 days (8) vs. 21 days (6) | Z = −2.4814, $p_{adj}$ = 0.130 | ns |
| qPCR | | 0 days (8) vs. 28 days (4) | Z = −3.1454, $p_{adj}$ = 0.0166 | * |
| *AmOARβ2* | | 7 days (8) vs. 14 days (7) | Z = −3.4897, $p_{adj}$ = 0.0048 | ** |
| DL | Dunns test | 7 days (8) vs. 21 days (6) | Z = −2.9841, $p_{adj}$ = 0.0284 | * |
| *Figure 3E* | | 7 days (8) vs. 28 days (4) | Z = −3.5887, $p_{adj}$ = 0.0033 | ** |
| | | 14 days (7) vs. 21 days (6) | Z = 0.3496, $p_{adj}$ = 1.0 | ns |
| | | 14 days (7) vs. 28 days (4) | Z = −0.6246, $p_{adj}$ = 1.0 | ns |
| | | 21 days (6) vs. 28 days (4) | Z = −0.9079, $p_{adj}$ = 1.0 | ns |

non-selective but potent octopamine receptor antagonist mianserin leads to hypothermia (*Figure 5A*, *Table 3*), while the effective tyramine receptor and $\alpha$ octopamine receptor antagonist yohimbine does not (*Figure 5B*, *Table 3*). Finally, alprenolol causes hypothermia too (*Figure 5C*, *Table 3*), whereas carvedilol and metoprolol did not have an observable effect on thermogenesis (*Table 3*).

## Downstream metabolic pathway analyses points to glycolysis

In a final experiment series, we investigated the signaling pathway downstream of octopamine receptors in more detail. Up this point, our results indicate the activation of β octopamine receptors, leading to an increase in cAMP concentration. This second messenger has the potential to activate protein kinase A (PKA). To test whether PKA is directly involved in the cellular pathway that enables thermogenesis, we used Rp-8-CPT-cAMPS which is a potent, metabolically stable and membrane-permeable inhibitor of PKA (*Dostmann et al., 1990*; *Gjertsen et al., 1995*). Rp-8-CPT-cAMPS negatively effects thermogenesis in both, nurse bees and forager bees (*Figure 6A*). Furthermore, we wanted to know whether octopamine release, which most likely activates PKA, could stimulate glycolysis. To test this hypothesis, we quantified pyruvate concentration in DL muscles after octopamine stimulation. Pyruvate is formed in the final step of glycolysis and its metabolites are further catabolized in the tricarboxylic acid cycle (*Zhang et al., 2019*). Pyruvate concentrations increase significantly after octopamine stimulation (*Figure 6B*). Finally, we observed that the *AmGAPDH* gene shows increased expression triggered by cold stress (*Figure 6C*). This gene encodes glyceraldehyde 3-phosphate dehydrogenase which converts glyceraldehyde 3-phosphate to D-glycerate 1,3-bisphosphate during glycolysis. A similar increase in *AmGAPDH* expression can be observed when the bees were treated with an octopamine injection in to the flight muscles instead of cold stress. (*Figure 6D*).

## Discussion

In this study, we hypothesized that octopamine has a critical role in the shivering thermogenesis of honeybees. An important prerequisite is that this monoamine can be used as a neurochemical substance at the flight muscles, which seems to be a conserved feature in insects (*Duch et al., 1999*; *Schlurmann and Hausen, 2003*; *Pauls et al., 2018*). We can demonstrate that octopamine is present in workerbee flight muscles by independent analysis methods (HPLC-ECD, antibody labeling). This is most likely delivered via flight muscle innervating neurons from the MMTG. Here, we can detect four octopaminergic cell clusters. Those are known to derive from a single median neuroblast at the posterior border of each segment of the developing neuroectoderm and are then displaced during the fusion of ganglia to the dorsal or the ventral surface (*Bräunig and Pflüger, 2001*). We postulate that the octopaminergic cells in each cluster we found are descendants of individual neuroblasts of their neuromere. The honeybee MMTG is formed by fusion of four neuromers (mesothorax, metathorax, first and second abdominal ganglia; *Markl, 1966*). Furthermore, the MMTG nerves IIN1 and

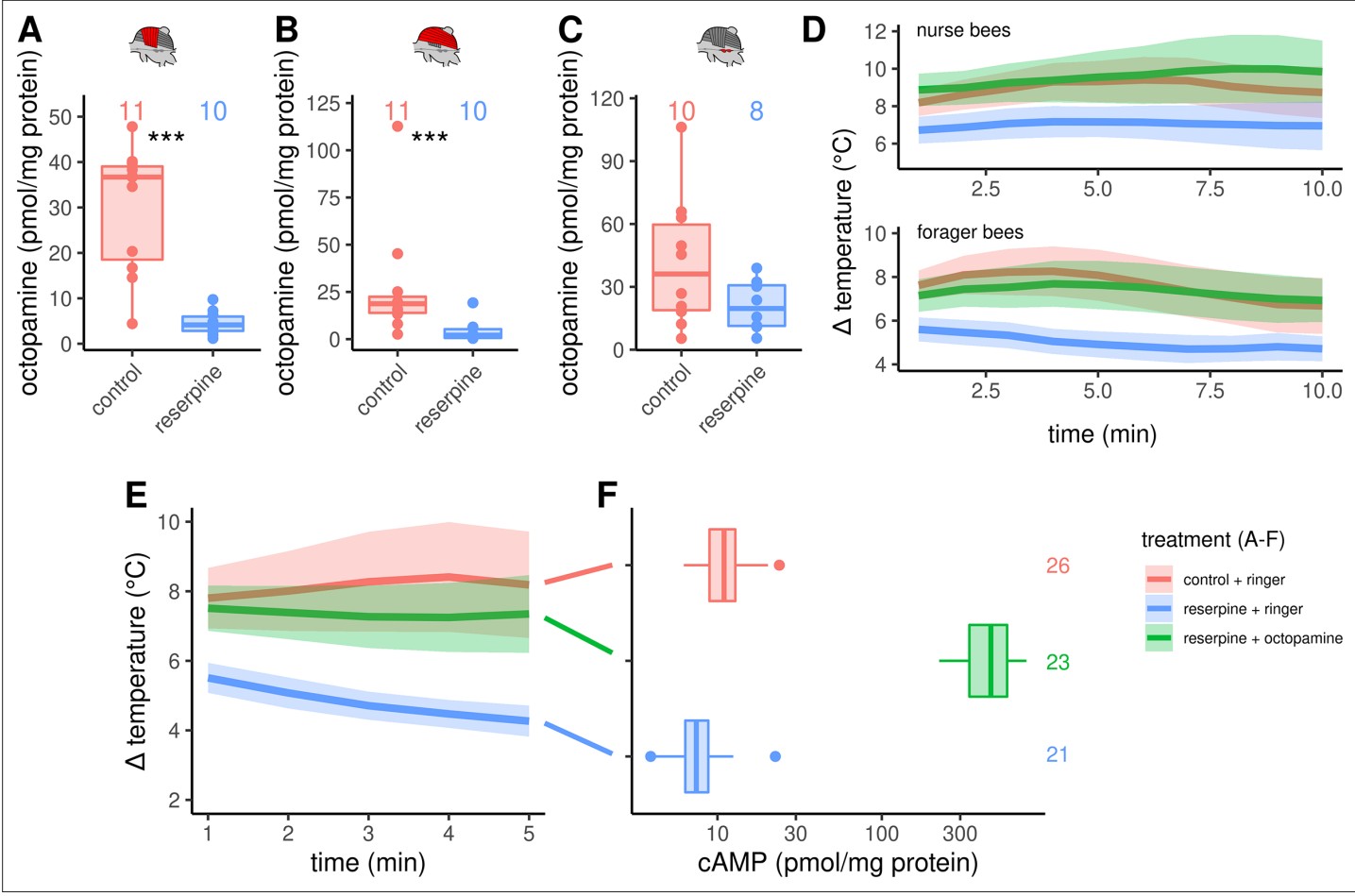

**Figure 4.** Octopaminergic control of honeybee thermogenesis. (**A–C**) Octopamine concentrations are decreased in DV and DL but not MMTG of reserpinized workerbees. Data are represented as boxplots. Shown is median ± IQR. Mann-Whitney *U* test, DV(A): W = 105, z = –3.70, p < 0.001; DL(B): W = 102, z = –3.37, p < 0.001; MMTG(C): W = 56, z = –0.94, p = 0.1728. (**D**) Reserpinized bees show hypothermia when compared with control. An injection of octopamine into the flight muscle helps the bees to recover, as no differences are observable between the control group and the recovered bees. The solid line represents the mean difference between $T_{THX}$ and $T_A$ and the shaded area represents the 95% confidence interval. For statistic see **Table 3**. (**E**) Similar experiment as in (**D**) but bees were frozen in liquid $N_2$ after 5 min for cAMP quantification. For statistic see **Table 3**. (**F**) The tissue cAMP concentrations in the flight muscles differ significantly due the treatment (Kruskal-Wallis test, $X^2$ = 52.636, df = 2, p < 0.001). Reserpinized bees has the lowest tissue cAMP concentrations in the flight muscles when compared with controls (Dunns test, Z = 2.6383, $p_{adj}$ = 0.025) and recovered bees (Z = 7.117, $p_{adj}$ = < 0.001). Controls also differ from the recovered bees (Z = –4.7998, $p_{adj}$ < 0.001). Data are represented as boxplots. Shown is median ± IQR.

The online version of this article includes the following figure supplement(s) for figure 4:

**Figure supplement 1.** The effect of reserpine on monoamine concentrations in DV, DL and MMTG.

**Figure supplement 2.** Time series of thermographic recordings of thoraces of selected workerbees.

**Figure supplement 3.** The effect of different monoamines on thermogenesis of reserpinized bees.

**Figure supplement 4.** The effect of reserpine on flight muscle cGMP concentrations.

IIN3 innervate DV and DL, respectively (**Markl, 1966**; **Pan, 1980**), while some of their neuronal structures contain octopamine. Finally, they reach DV and DL as octopaminergic varicosities suggest. We conclude, that octopaminergic neurons from the MMTG directly innervate the flight muscles and therefore influence thermogenesis.

If this is true, octopamine should be detectable at comparable concentrations in the flight muscles of workerbees capable of thermogenesis. Indeed, we found no differences in bees with ages ranging

**Table 3.** Statistical analysis of the thermogenesis dependent on the pharmacological treatment.

c = control, *r* = reserpine, ATS = ANOVA type statistic, ns = not significant.

| Experiment | Groups (n) | ATS | Df | p | |
|---|---|---|---|---|---|
| | | 9.3635 | 1.9854 | 0.00009 | *** |
| Reserpine Nurse bees *Figure 4D* | c + ringer (21) vs. *r* + ringer(23) | 13.9618 | 1.0 | 0.0002 | *** |
| | c + ringer (21) vs. *r* + octopamine (23) | 0.0952 | 1.0 | 0.7577 | ns |
| | *r* + ringer(23) vs. *r* + octopamine (23) | 14.2223 | 1.0 | 0.0002 | *** |
| | | 14.5704 | 1.9437 | 0.0000006 | *** |
| Reserpine Forager bees *Figure 4D* | c + ringer (29) vs. *r* + ringer(28) | 126.5492 | 1.0000 | 0.0000003 | *** |
| | c + ringer (29) vs. *r* + octopamine (29) | 0.0753 | 1.0 | 0.7838 | ns |
| | *r* + ringer(28) vs. *r* + octopamine (29) | 21.1833 | 1.0000 | 0.000004 | *** |
| | | 22.8759 | 1.8981 | 0,0000000003 | *** |
| Reserpine cAMP Quantification *Figure 4E* | c + ringer (26) vs. *r* + ringer(21) | 39.9913 | 1.0000 | 0.0000000003 | *** |
| | c + ringer (26) vs. *r* + octopamine (23) | 0.1155 | 1.0 | 0.734 | ns |
| | *r* + ringer(21) vs. *r* + octopamine (23) | 37.3015 | 1.0000 | 0.000000001 | *** |
| Mianserin Nurse bees Forager bees *Figure 5A* | control (30) vs. mianserin (30) | 9.2737 | 1.0000 | 0.0023 | ** |
| | control (30) vs. mianserin (30) | 8.4638 | 1.0000 | 0.0036 | ** |
| Yohimbine Nurse bees Forager bees *Figure 5B* | control (30) vs. yohimbine (30) | 0.8011 | 1.0000 | 0.3708 | ns |
| | control (32) vs. yohimbine (33) | 0.0584 | 1.0000 | 0.8091 | ns |
| Alprenolol Nurse bees Forager bees *Figure 5C* | control (30) vs. alprenolol (30) | 7.5516 | 1.0000 | 0.0059 | ** |
| | control (34) vs. alprenolol (33) | 10.9721 | 1.0000 | 0.0009 | *** |
| Carvedilol Nurse bees Forager bees | control (30) vs. carvedilol (30) | 0.1235 | 1.0000 | 0.7252 | ns |
| | control (36) vs. carvedilol (34) | 0.2650 | 1.0000 | 0.6067 | ns |
| Metoprolol Nurse bees Forager bees | control (30) vs. metoprolol (30) | 0.1031 | 1.0000 | 0.7481 | ns |
| | control (36) vs. metoprolol (36) | 0.2029 | 1.0000 | 0.6524 | ns |
| Rp-8-CPT-cAMPS Nurse bees Forager bees *Figure 6A* | control (25) vs. Rp-8-CPT-cAMPS (23) | 4.062 | 1.0000 | 0.044 | * |
| | control (15) vs. Rp-8-CPT-cAMPS (14) | 27.7439 | 1.0000 | 0.0000001 | *** |

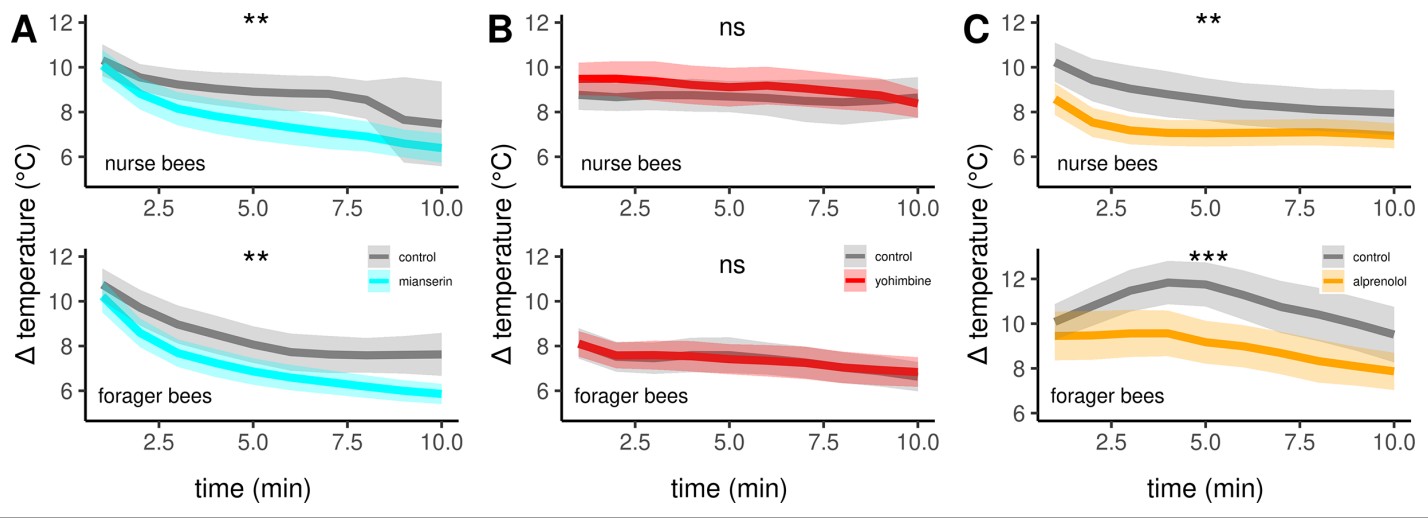

**Figure 5.** The effects of different antagonists on workerbee thermogenesis. Mianserin (**A**) and alprenolol (**C**) cause hypothermia in workerbees but not yohimbine (**B**). The solid line represents the mean difference between $T_{THX}$ and $T_A$ and the shaded area represents the 95% confidence interval. For statistic see **Table 3**.

from 7 days up to 4 weeks. They are all similarly engaged in active heat production independent of their actual task within the colony (**Stabentheiner et al., 2010**). In contrast, newly emerged bees, which are not capable of heat production (**Harrison, 1987**; **Stabentheiner et al., 2010**), have significant lower flight muscle octopamine concentrations. It remains uncertain whether there is a causal relationship between the low octopamine concentrations and the absence of thermogenesis in newly emerged bees or whether this observation is merely a correlation. Several factors could be responsible, such as incomplete differentiation of flight muscle tissues (**Roberts and Elekonich, 2005**; **Correa-Fernandez and Cruz-Landim, 2010**).

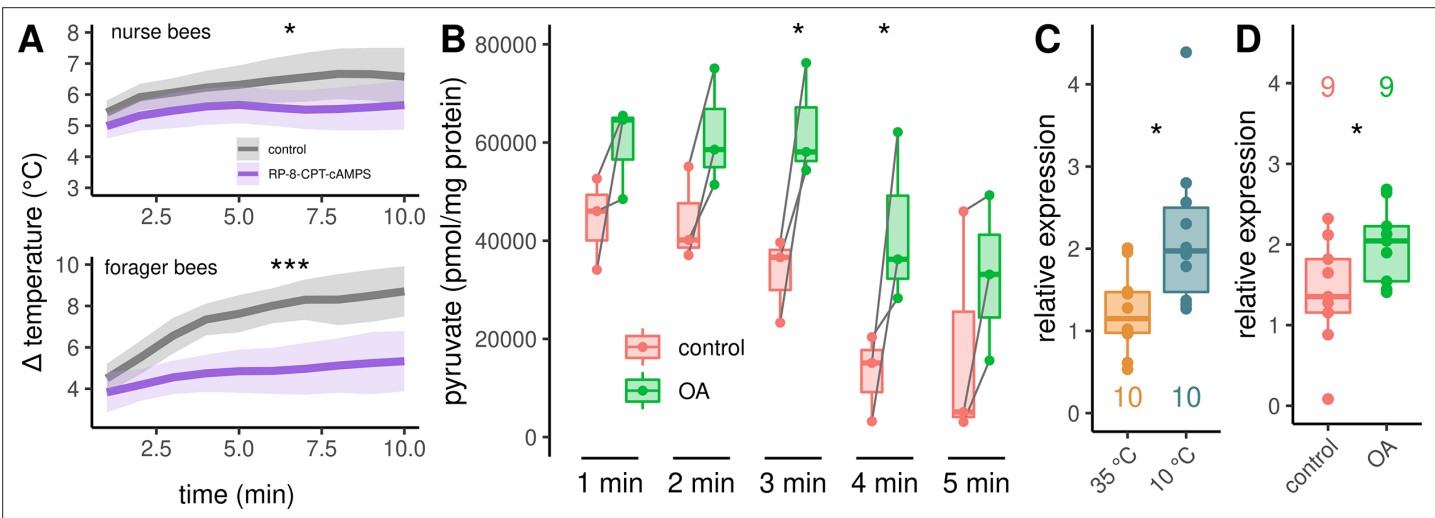

**Figure 6.** Analysis of the downstream pathway. (**A**) The PKA inhibitor Rp-8-CPT-cAMPS causes hypothermia in nurse bees and forager bees. The solid line represents the mean difference between $T_{THX}$ and $T_A$ and the shaded area represents the 95% confidence interval. For statistic see **Table 3**. (**B**) DL muscles were separated into two mirror-similar parts and treated differently. Bath application of octopamine (**B**) leads to an difference in the mean pyruvate concentration when compared with control (two-way RM ANOVA, F(1)=38.28, p < 0.001). The simple main effect of treatment becomes significant after 3 and 4 min (Sidaks multiple comparisons test, 3 min: p = 0.016, 4 min: p = 0.017). Shown is median ± IQR. Data points of the same individual are connected by gray lines. (**C–D**) *AmGAPDH* expression in DV and DL is upregulated due to cold stress (C, Mann-Whitney *U* test, W = 83, z = –2.24, p = 0.01261). This result can be mimicked by an injection of octopamine directly into the flight muscles (D, Mann-Whitney *U* test, W = 18, z = –1.68, p = 0.04694). Shown is median ± IQR.

We determined *AmOARα1* and *AmOARβ2* as the predominant octopamine receptor genes expressed in the flight muscle, and their expression is detectable across age. The relative expression of both genes is higher in older bees, but at the same time a huge inter-individual variation is detectable. This might reflect differential demands to muscle activity in the context of the age-dependent task allocation and its neurochemical control. Workerbees perform very different tasks as a function of their age (*Seeley, 1995*). Yet, they are all similarly engaged in heat production if they are older than two days (*Stabentheiner et al., 2010*). Besides flight and thermogenesis another important function of the flight muscles is fanning for cooling purposes (*Hess, 1926*; *Hazelhoff, 1954*; *Simpson, 1961*) and octopamine is known to increase the probability of fanning when fed to workerbees together with tyramine (*Cook et al., 2017*). The two genes *AmOARα1* and *AmOARβ2* encode the octopamine receptor proteins AmOARα1 (*Grohmann et al., 2003*) and AmOARβ2 (*Balfanz et al., 2014*), respectively. We assume, that both receptors can receive and forward the signal mediated by an octopamine release at the flight muscles. Until now, we did not know in which situations this occurs and what specific role the corresponding receptors might have in this process.

Our reserpine experiments solve this problem, because it makes octopamine no longer usable at the flight muscle. As direct consequence, we observe hypothermia. Moreover, if we supply the system with octopamine again we can restore heat generation. We conclude that octopamine signaling is necessary for honeybee thermogenesis. This interpretation is supported by the fact that the potent octopamine receptor antagonist mianserin (*Grohmann et al., 2003*; *Balfanz et al., 2014*; *Blenau et al., 2020*) causes hypothermia, too. Moreover, our cAMP quantification result suggests that at least one β octopamine receptor subtype mediates the octopamine signal in the service of thermogenesis. The decreased octopamine availability in the flight muscles of reserpinezed bees likely causes the loss of octopamine release if necessary. In the end, this results in a reduction of octopamine receptor activation events. In the case of β octopamine receptors, consequently, no cAMP is produced. Indeed, we observe a decrease in tissue cAMP concentrations in combination with reserpine induced hypothermia. Octopamine-induced reversal of this effect is accompanied by a tremendous increase in tissue cAMP concentrations. Unfortunately, honeybee cAMP concentrations from muscle tissues are not available, but our results are consistent with analysis in locust flight muscle (*Baines et al., 1990*; *Lange and Nykamp, 1996*). Furthermore, the lack of an octopamine effect on cGMP concentrations and the absence of the other cyclic nucleotides clearly suggests that mACs mediate the observed octopamine effects. *Hasan et al., 2014* could show that mAC activation leads to exclusive cAMP increase. Our results strongly suggests that β octopamine receptor activation is necessary for honeybee thermogenesis, since these receptors are known to be positively coupled mACs (*Balfanz et al., 2014*). Our explanation again receives support by pharmacological thermography. Due to the lack of subtype-specific octopamine receptor antagonists, we made use of well-established adrenoceptor antagonists. Deuterostome adrenoceptor and arthropoda octopamine receptors are very closely related (*Roeder, 2005*; *Spindler et al., 2013*; *Fuchs et al., 2014*; *Hochman, 2015*; *Roeder, 2020*), which also applies to receptor subtypes as supported by phylogenetic analyses (*Qi et al., 2017*). The reserpine and mianserin effects described above can be mimicked by alprenolol (*Figure 5G*). This antagonist is active at both, β1 and β2 adrenoceptors (*Åblad et al., 1973*; *Åblad et al., 1972*). Therefore, it represents a putative antagonist of AmOARβ1 and AmOARβ2 and was already used in insects in other studies (*Belzunces et al., 1996*; *Cossío-Bayúgar et al., 2012*). Contrastly, carvedilol and metoprolol did not cause any effect. Carvedilol antagonizes preferably α1 and β1 adrenoceptors (*Hansson and Himmelmann, 1998*), whereas metoprolol antagonizes βone adrenoceptors in the human heart (*Benfield et al., 1986*). We assume that both substances antagonize the corresponding octopamine receptors. Several studies show that metoprolol is effective in species belonging to all major protostome phyla (*Dzialowski et al., 2006*; *Spindler et al., 2013*; *Jungmann et al., 2017*; *Buchberger et al., 2018*). However, an expansion of the pharmacological profiles of honeybee octopamine receptors (*Grohmann et al., 2003*; *Balfanz et al., 2014*; *Blenau et al., 2020*) is needed to confirm whether the compounds we used actually antagonize the desired receptor proteins. Combining the information stated above with our results that alprenolol causes hypothermia but not yohimbine, which does not antagonize honeybee β octopamine receptors (*Balfanz et al., 2014*; *Kovac et al., 2009*), further supports the hypothesis that at least one β octopamine receptor subtype is crucially involved in honeybee thermogenesis. Since *AmOARβ2* is predominantly expressed in the flight muscles (when compared with *AmOARβ1* and A*mOARβ3/4*), AmOARβ2 is the most promising candidate. This

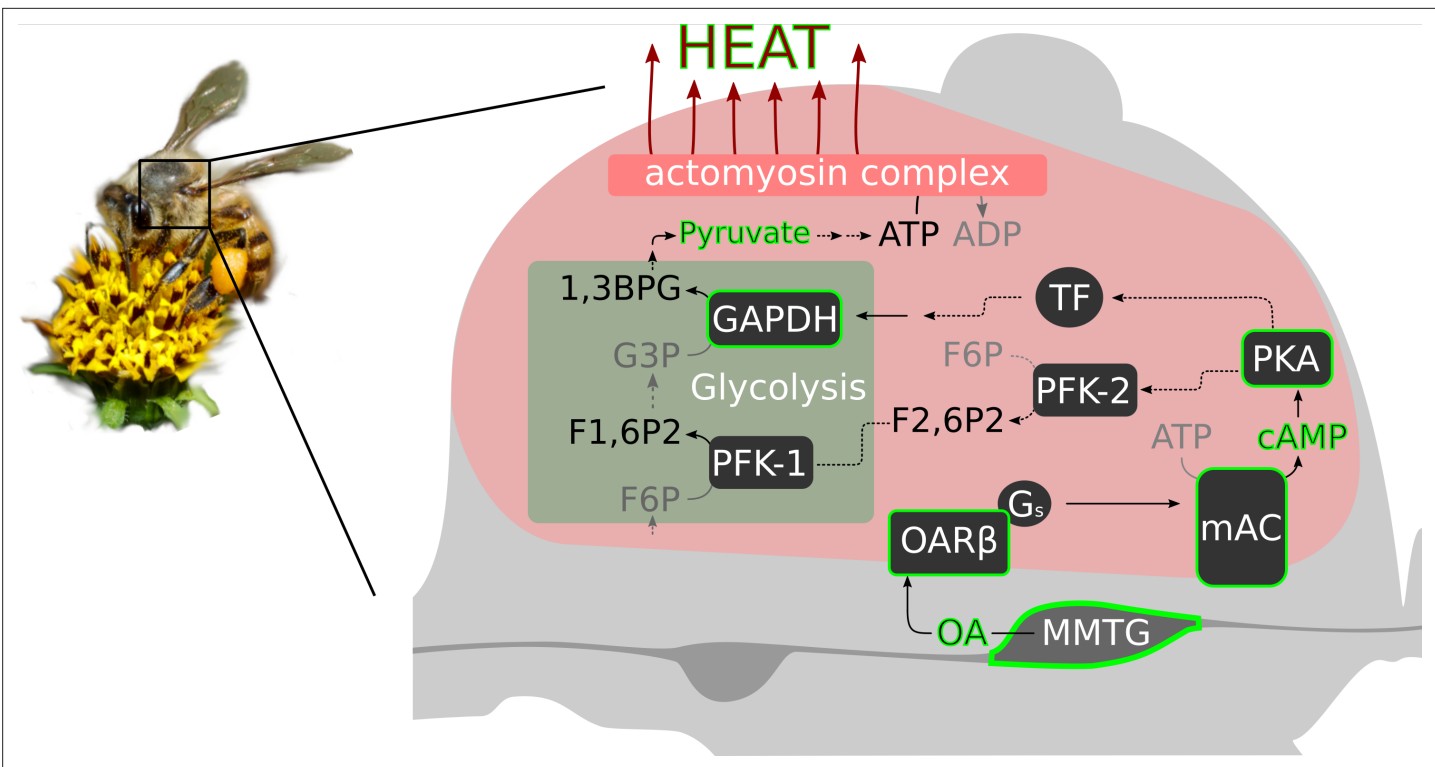

**Figure 7.** Octopamine and honeybee thermogenesis. The scheme summarizes our findings, with the solid lines and green borders representing interpretations supported by our results and the dashed lines representing hypothetical pathways. Muscle innervating neurons in the MMTG release octopamine (OA) directly to the flight muscles. By this, AmOARβ two receptors are activated which in turn activate the membrane-bound adenylyl cyclase (mAC) via $G_s$ proteins. The resulting increase in the intracellular cAMP concentration leads to the activation of Proteinkinase A (PKA) which phosphorylates and by this activates phosphofructokinase 2 (PFK-2). Consequently, this enzyme produces fructose-2,6-bisphosphate (F2,6P2) which increases the activity of phosphofructokinase 1 (PFK-1). An alternative pathway is the PKA mediated activation of transcription factors (TF) which might enhance expression of *GAPDH* which encodes glyceraldehyde 3-phosphate dehydrogenase (GAPDH). This enzyme converts glyceraldehyde 3-phosphate (G3P) into 1,3-bisphosphoglyceric acid (1,3BPG). All together, this increases the glycolysis rate so that a greater amount of pyruvate is available for ATP production. Finally, heat is generated in the actomyosin complex under ATP consumption.

assumption is supported by studies in mammals showing the predominant expression of the βtwo adrenergic receptor in skeletal muscle tissue, which is a similar receptor subtype (*Liggett et al., 1988*; *Kim et al., 1991*).

Our PCR analysis further revealed the prevalent expression of *AmOARα1*. However, yohimbine does not cause hypothermia. This substance was shown to bind and antagonize αone octopamine receptors receptors in a wide range of insects (*Bischof and Enan, 2004*; *Enan, 2005*; *Ohtani et al., 2006*; *Huang et al., 2010*). Thus, we hypothesize that this receptor is not in the service of thermogenesis.

Tyramine is also capable to reverse the reserpine induced hypothermia. However, we could observe neither tyramine nor any tyramine receptor gene expression in the flight muscles. One might argue that the tyramine effect is mediated via tyramine receptors that are expressed in the MMTG. In that case, the potent tyramine receptor antagonist yohimbine (*Ohta et al., 2003*; *Fussnecker et al., 2006*; *Reim et al., 2017*) should have an effect on thermogenesis, but this is not the case. Based on our results and the fact that tyramine is able to activate octopamine receptors (*Grohmann et al., 2003*; *Balfanz et al., 2014*; *Blenau et al., 2020*), we classify this tyramine effect as artificial and physiologically not relevant.

The data of our study supports the hypothesis that octopaminergic signaling in the flight muscle is necessary for honeybee thermogenesis. Most likely, this monoamine acts directly at the indirect flight muscles via the activation of β octopamine receptors. We speculate, that their role is to boost glycolysis (see scheme *Figure 7*). Cold stress will induce an octopamine release directly at the flight muscles. The subsequent β octopamine receptor mediated generation of cAMP will activate proteinkinase A (PKA, *Müller, 2000*). That PKA is in service of thermogenesis is supported by our experiments in which

bees are hypothermic as a result of PKA inhibition. PKA in turn might phosphorylates and activates phosphofructokinase 2 (PFK-2), which is the enzyme that produces fructose-2,6-bisphosphate (F2,6P$_2$, *Rider et al., 2004*). F2,6P$_2$ is an activity increasing modulator of phosphofructokinase 1 (PFK-1, *Hue and Rider, 1987*; *Bartrons et al., 2018*). The PFK-1 mediated phosphorylation of fructose-6-phosphate (F6P) to fructose-1,6-bisphosphate (F1,6P$_2$) is a key step in glycolysis, at its end ATP is provided (*Fothergill-Gilmore and Michels, 1993*). Finally, heat is generated by the hydrolysis of ATP at the actomyosin complex (*Zhang and Feng, 2016*). Our pyruvate quantification results support this hypothesis. We can detect higher quantities of the glycolysis final product after octopamine stimulation. Another possibility is that PKA is involved in the activation of certain transcription factors. As a consequence, the expression of genes of important glycolysis enzymes may be enhanced. Here, we provide the *AmGAPDH* gene as one example whose gene product is essential in glycolysis (*Burke et al., 1996*). Its expression can be increased by both cold stress and octopamine injection. The alternative futile cycle (*Newsholme et al., 1972*), which is based on high fructose-1,6-bisphosphatase (FbPase) activity in certain bumblebee species, must be doubted, at least for honeybees. Honeybees and many other bumblebee species have comparable low FbPase activity (*Newsholme and Crabtree, 1970*; *Staples et al., 2004*) and FbPase-PFK cycling rates are not sufficient for heat production (*Clark et al., 1973*; *Kammer and Heinrich, 1978*; *Newsholme and Crabtree, 1976*). Our hypothetical cascade is supported by the results of other studies. F2,6P$_2$ levels increase in locust flight muscles due to octopamine stimulation (*Blau and Wegener, 1994*; *Blau et al., 1994*) and by this controls the rate of carbohydrate oxidation in flight muscles (*Wegener, 1996*). In mammals, adrenaline stimulates increasing F2,6P$_2$ levels and thus glycolysis (*Narabayashi et al., 1985*). This effect is achieved by β adrenoreceptor activation followed by stimulation of PKA (*Rider et al., 2004*). Chronic exercise causes stereotypical adaptations in several tissues of *Drosophila melanogaster*, which requires the activation of octopaminergic neurons (*Sujkowski et al., 2017*). In muscles, those effects are dependent on the activation of β octopamine receptors (*Sujkowski et al., 2020*). If cold stress becomes chronic, such as in cold climate at high altitude or during winter, there will probably be a similar pattern in honeybees. It is conceivable that the octopaminergic system in the flight muscles is permanently active to enable persistent heat production. If this system is compromised, it will endanger the survival of the colony due to the lost of individually performed heating, which enables foraging, breeding, and diverse defense mechanisms (*Himmer, 1932*; *Weiss, 1962*; *Starks et al., 2000*; *Bujok et al., 2002*; *Stabentheiner et al., 2003*; *Tautz et al., 2003*; *Ken et al., 2005*; *Baracchi et al., 2010*; *Campbell et al., 2010*; *Wang et al., 2016*). This may explain the enormous selective pressure on β octopamine receptor genes (*Wallberg et al., 2017*). Issues to be addressed are how the octopaminergic system responds to cold stress. But also heat stress, and in this context adaptations to warm climate in the course of climate change can become very important. With our important contribution to the understanding of thermogenesis in honeybees we provide a solid basis to analyze these issues.

## Materials and methods

### Animals

Honeybee workers (*Apis mellifera carnica*) were collected from colonies of the department next to the Biocenter at the University of Würzburg, Germany. We declared bees that returned to the hive with pollen loads on their hind legs as forager bees. As nurse bees, we defined bees, that were sitting on a brood comb and were actively heating (thorax temperature, $T_{THX} \geq 32$ °C). $T_{THX}$ was monitored with a portable thermographic camera (FLIR E6, FLIR, Wilsonville, USA). Pollen forager were collected for the gene expression analysis from the same hives and were immediately flash-frozen in liquid nitrogen and stored at –80 °C. For the age-series analysis (monoamine quantification, gene expression analysis), a queen was caged on a brood comb for 3 days. Shortly before the bees started to emerge, we transferred the brood comb into an incubator (34 °C, RH = 65 %). The first group (0-day-old bees) consisted of newly hatched bees and were collected directly from the brood comb. The remaining newly hatched bees were color-marked and then inserted into a standard hive. Those bees were collected from the hive after 7, 14, 21, and 28 days, respectively. For the *AmGAPDH* expression analysis, 7-day-old age-marked bees were collected from a hive and distributed equally into two identical cages. For the cold stress experiment, one cage was placed in an incubator at 10 °C for 120 min, while the other served as a control (120 min, 34 °C). For the octopamine injection experiment, bees of the

control group receive an injection of saline solution (270 mM sodium chloride, 3.2 mM potassium chloride, 1.2 mM calcium chloride, 10 mM magnesium chloride, 10 mM 3-(N-morpholino) propanesulfonic acid, pH = 7.4; *Erber and Kloppenburg, 1995*) into their flight muscles. The test group was injected with octopamine (0.01 M in saline). Subsequently, both groups were incubated for 120 min at 34 °C. All collected bees (expression analysis, monoamine quantification) were immediately flash-frozen in liquid nitrogen and subsequently stored at –80 °C.

## Immunohistochemistry

For octopamine immunolabeling, we used a polyclonal rabbit anti-octopamine antibody (IS1033, ImmuSmol, Bordeaux, France) together with the STAINperfect immunostaining kit A (SP-A-1000, ImmuSmol, Bordeaux, France). We have analyzed ten individual MMTGs in three independent experiments for frontal sections and additionally three individual MMTGs for sagital sections. Four individual DVs and DLs, respectively, were analyzed in two independent experiments. Due to non optimal tissue permeability, we have slightly adopted the manufacturers protocol for whole mount preparations to perform analysis with vibratom sections. In brief, tissues (MMTG, flight muscles) were micro-dissected and subsequently fixed in fixation buffer for 3 hr at 4 °C while shaking. Afterwards, the fixed tissues were washed five times for 30 min with *Wash Solution 1*, embedded in 5% (w/v) agarose and were cut into 100-µm-thick sections. Then, the tissue sections were treated consecutively: 1 hr in *Permeabilization Solution* at RT followed by two times *Wash Solution 1* for 3 min, 1 hr in *Stabilization Solution* followed by three times *Wash Solution 1* for 3 min, and 1 hr in *Saturation Solution* at RT. Afterwards, the *Saturation Solution* was replaced by the primary antibody (1:500, in *Antibody Diluent*) and the tissue sections were incubated at 4 °C while shaking for at least for 72 hr. After five times washing cycles with *Wash Solution 2* for 30 min at RT the secondary antibody (1:200 in *Antibody Diluent*, goat anti-rabbit Alexa Fluor 568; Molecular Probes, Eugene, USA) was applied for 24 hr (4 °C). After the final washing with *Wash Solution two* and *Wash Solution 3* (both 3 times for 30 min at RT) the slices were mounted in 80% Glycerol (in *Wash Solution 3*) on microscope slides. Preparations were imaged by confocal laser scanning microscopy using a Leica TCS SP2 AOBS (Leica Microsystems AG, Wetzlar, Germany). HC PL APO objective lenses (10 x/0.4 NA imm; 20 x/ 0.7 NA imm and 63 x/1.20 NA imm) with additional digital zoom were used for image acquisition. ImageJ (1.53 c, *Schindelin et al., 2012*) was used to process images (maximum intensity projection, optimization for brightness and contrast) and Inkscape (1.1, *Inkscape Developer Team, 2021*) was used to arrange images into figures. MMTG nerve terminology is based on the nomenclature used by *Markl, 1966*.

## Monoamine quantification

The DV and DL were dissected under liquid nitrogen. Afterwards, we thawed the remaining thoracic tissue in ice-cold ethanol to immediately dissect the MMTG. The separated tissues were kept at –80 °C until extraction. For high-performance liquid chromatography (HPLC) analysis of the monoamines we used a slightly modified protocol as described by *Cook et al., 2017*. For extraction, 120 µL (DV, DL) or 60 µL (MMTG) of extraction solution (10.0 pg/µL 3,4-dihydroxy-benzylamine (DHBA) in 0.2 M perchloric acid) was added in the first step. After a short centrifugation (21,130 g, 2 min, 0 °C) the tissues were disintegrated via sonication (10 min, 0 °C), followed by an incubation (20 min, 0 °C). After a final centrifugation (21,130 g, 14 min, 0 °C), the supernatant was analyzed via HPLC-ECD (Thermo Fisher Scientific, Waltham, USA) and the pellet was stored at –80 °C for protein quantification. A 3 µm reverse phase column (BDS-Hypersil-C18, 150 × 3 mm, pore size 130 Å, Thermo Fisher Scientific, Waltham, USA) and an ECD-3000RS configuration with two coulometric cells (6011RS ultra-analytical cell, Thermo Fisher Scientific, Waltham, USA) were connected to a biocompatible Dionex Ultimate 3,000 UHPLC focused (Thermo Fisher Scientific, Waltham, USA). The mobile phase contained 15% (v/v) methanol, 15% (v/v) acetonitrile, 85 mM sodium phosphate monobasic, 1.75 mM sodium dodecyl sulfate, 0.5 mM sodium citrate and ultrapure water. Phosphoric acid was used for accurate pH adjustment (pH 5.6 ± 0.01). We used a flow rate of 0.5 mL/min. Two detector channels were connected in series with working potentials of 425 mV (DHBA, dopamine, serotonin) and 800 mV (octopamine, tyramine), respectively. Quantification was performed via an external calibration. The raw data analysis was carried out with the program Chromeleon (7.2.10, Thermo Fisher Scientific, Waltham, USA).

## Quantitative analysis of cyclic nucleotides

Individual flight muscle tissues were dissected under liquid nitrogen. Individual DV and DL were pooled and 800 µL homogenization buffer (40% (v/v) acetonitrile, 40% (v/v) methanol, 20% (v/v) $H_2O$) was added and homogenized as described above. Samples were incubated at 95 °C for 10 min and then stored in the freezer (–80 °C) until further processing. After centrifugation (10 min, 21,130 g), the supernatant was transferred to mass spectroscopic analysis (HPLC-MS) as described by *Beste et al., 2012*. The residual pellet was used for the protein quantification.

## Pyruvate quantification

Workerbees were killed by decapitating and then the intact DL muscle was carefully dissected and separated into mirror-identical parts. Subsequently, both parts were incubated with different solutions using bath application. One part was treated with saline solution whereas the other part was treated with 0.01 M octopamine (in saline). After flash freezing in liquid nitrogen pyruvate was quantified using the pyruvate assay kit (MAK071, Sigma Aldrich). The muscles were homogenized in 100 µL Pyruvate Assay Buffer and in a tissue mill at 35 Hz for 3 min. After centrifugation (10 min, 21,130 g), 25 µL of the supernatant were used per reaction. Each reaction setup (50 µL) additionally contained 23 µL Pyruvate Assay Buffer, 1 µL Pyruvate Probe Solution and 1 µL Pyruvate Enzyme Mix. After incubation at room temperature for 30 min the absorption at 570 nm was measured for each sample and each external calibrator (0, 2, 4, 6, 8, 10 nmol per reaction).

## Protein quantification

To compensate possible differences in the accuracy of tissue dissection for the HPLC-ECD, HPLC-MS and the pyruvate quantification analysis, we additionally measured the protein content in the samples after Bradford (*Fic et al., 2010*) and normalized amine or cyclic nucleotide concentration to protein content. The pellet (see above) was resuspended in 120 µL (HPLC-ECD: DV, DL), 30 µL (HPLC-ECD: MMTG), or 500 µL (HPLC-MS: DV+ DL) 0.2 M NaOH. After an incubation (15 min, 0 °C), the insoluble material was sedimented (9391 g, 5 min). Finally, 2 µL (HPLC-ECD: DV, DL), 10 µL (HPLC-ECD: MMTG), or 2,5 µL (HPLC-MS: DV+ DL) of the supernatant were transferred into a final volume of 1 mL 1 x ROTI-Nanoquant solution (Carl Roth, Karlsruhe, Germany). All samples and the external calibrator (1, 2, 3, 5, 10, 20 µg/mL Albumin Fraction V, Carl Roth, Karlsruhe, Germany) were analyzed with a plate reader (Infinite 200 Pro, Tecan, Männedorf, Switzerland).

## Gene expression analysis

Individual flight muscle tissues were dissected under liquid nitrogen. For the MMTG, we have used RNAlater ICE (Thermo Fisher Scientific, Waltham, USA) to prevent RNA degradation during the dissections. The GenUP Total RNA Kit (biotechrabbit, Henningsdorf, Germany) was used to extract total RNA following the standard protocol provided by the manufacturer including an extra DNase I digestion step. After binding of the RNA to the Mini Filter RNA, we added a 50 µL DNase mix containing 30 U RNase-free DNase I (Lucigen Corporation, Middleton, USA) together with the appropriate buffer and incubated for 15 min at room temperature. For the polymerase chain reaction (PCR) experiment, we pooled total RNA from one individual of each age (7, 14, 21 and 28-day-old bees) per tissue (brain, MMTG, DV, DL). 400 ng total RNA of each tissue were used for cDNA synthesis using the Biozym cDNA Synthesis Kit (Biozym, Hessisch Oldendorf, Germany). The cDNAs were then analyzed in 20 µL PCR reactions (1 µL cDNA, 8.2 µL $H_2O$, 10 µL 2 x qPCR S'Green BlueMix (Biozym, Hessisch Oldendorf, Germany)), 0.4 µL of each primer (0.2 µM) using the following protocol: 95 °C for 2 min and 35 cycles at 95 °C for 5 s and 30 °C for 30 s. Finally, 10 µL for each PCR reaction was analyzed on a 1.5% agarose gel. For the qPCR experiments, we used individual total RNA per tissue. Here, for each sample 70 ng (DV) and 30 ng (DL) RNA were used. All cDNA synthesis reactions were performed with the Biozym cDNA Synthesis Kit (Biozym, Hessisch Oldendorf, Germany). PCR triplicates of each cDNA (5 µL) were analyzed in a qPCR on a Rotor-Gene Q (Qiagen, Hilden, Germany) in a total reaction volume of 20 µL. Every reaction contains 4.2 µL $H_2O$, 10 µL 2 x qPCR S'Green BlueMix (Biozym, Hessisch Oldendorf, Germany), 0.4 µL of each primer (0.2 µM) and 5 µL cDNA. Finally, octopamine receptor gene expression was determined relative to the reference genes *AmGAPDH* and *AmRPL10* using the R package 'EasypcR' (v1.1.3) which uses the algorithm published by *Hellemans et al., 2007*. For the *AmGAPDH* relative expression analysis *AmRPL32* and *AmRPL19* served as reference genes.

## Pharmacological thermography

For the reserpine experiments, forager bees and nurse bees were collected as described above. The bees were kept and fed in equal proportions in two adjacent cages (34 °C, RH = 65 %) for 3 days. The reserpine group was fed with 500 µM reserpine solution (in 30% sucrose solution) ad libitum and the control group with 30% sucrose only. To enhance the solubility, the reserpine was pre-dissolved in acetone. For the experiments with receptor antagonists, the day before each measuring day, 20 bees were collected from the same hive and kept overnight in a cage at 34 °C (RH = 65 %). In the incubator, the bees were fed ad libitum with 30% sucrose solution. All injection solutions were freshly prepared every experimental day. All biogenic amines (Sigma-Aldrich), receptor antagonists (Sigma-Aldrich) or Rp-8-CPT-cAMPS (Biolog) were used in a concentration of 0.01 M in saline solution (see above). For solubility reasons, a 10:1 volume mixture of saline solution and dimethyl sulfoxide was used for carvedilol instead of pure buffer. Each bee was immobilized on ice until no more movement could be detected. The thorax was then punctured centrally to inject 1.0 µL testing solution using a 10.0 µL Hamilton syringe. Directly before the start of every measurement, the control group received an injection of the pure saline solution and the treatment group an injection of 0.01 M of the biogenic amine or the respective antagonist directly into their flight muscles. To enable optimal conditions for thermogenesis and thermographic recordings, we adapted the method of a tethered animal that walks upon a treadmill (*Moore et al., 2014*). This allows the bee to seemingly move freely, while at the same time the camera always monitors the same area of the bees thorax. This setup was located inside an incubator (18.5 °C, RH = 65 %) together with a thermographic camera (FLIR A65 camera, lens: 45°, $f$ = 13 mm, FLIR, Wilsonville, USA). A thermal imaging video with 30 frames/min was recorded of each bee over 10 min. We converted the thermographic videos using the R package Thermimage (4.1.2, *Tattersall, 2020*) to subsequently read out the thoracic temperatures with ImageJ (1.53 c, *Schindelin et al., 2012*).

## Statistical analysis

All statistical analyses were performed using R (4.0.4 including 'stats', *R Development Core Team, 2020*) and the R packages 'rstatix' (0.7.0, *Kassambara, 2020b*) and 'FSA' (0.9.1, *Ogle et al., 2021*). We performed a Shapiro-Wilk test to check the data for normality distribution. Since most data subsets did not display a normal distribution, we analyzed the data using either the Mann-Whitney $U$ test or the Kruskal-Wallis test followed by Dunns post hoc analysis if significant differences were observable. For the statistical analysis of the pharmacological thermography experiments, we calculated the mean value per min for $T_{THX}$ and $T_A$, respectively. Afterwards, the Δ temperature ($T_{THX} — T_A$) for the total time span of the experiment (five or 10 min) was subjected to nonparametric analysis of longitudinal data using a F1 LD F1 model of the R package 'nparLD' (2.1 *Noguchi et al., 2012*). Visualization of the data was performed with the R packages 'ggplot2' (3.3.3, *Wickham, 2016*), 'ggpubr' (0.4.0, *Kassambara, 2020a*), 'png' (0.1–7, *Urbanek, 2013*), 'cowplot' (1.1.1, *Wilke, 2019*), and 'magick' (2.7.0, *Wilke, 2019*).

## Acknowledgements

We thank Dirk Ahrens for beekeeping; Annette Garbe, Anna Senker, Feriba Ghanbari, Karin Möller, Lena Wolf, Linnéa Jürgensen, Saskia Delac and Valentina Vey for technical assistance during the experiments; Flavio Roces, Johannes Spaethe, Marcus Gutmann, Petra Högger, Ricarda Scheiner and Robin Grob for supplying technical equipment and Michael G.K. Brunk for his help with ImageJ. Usage of the Leica SP2 confocal Laser Scanning Microscope was made possible by the the Deutsche Forschungsgemeinschaft (DFG, German Research Foundation) - HBFG-133–612. A special thanks goes to Christian Wegener, Jens Schlossmann, Anna Stöckl and Basil el Jundi for fruitful discussions and/or reviewing earlier versions of the manuscript. This publication was supported by the Open Access Publication Fund of the University of Wuerzburg.

## Additional information

### Funding

| Funder | Grant reference number | Author |
| --- | --- | --- |
| Deutsche Forschungsgemeinschaft | TH2264/2-1 | Markus Thamm |

The funders had no role in study design, data collection and interpretation, or the decision to submit the work for publication.

### Author contributions

Sinan Kaya-Zeeb, Conceptualization, Formal analysis, Investigation, Methodology, Visualization, Writing - original draft, Writing - review and editing; Lorenz Engelmayer, Mara Straßburger, Investigation, Writing - review and editing; Jasmin Bayer, Investigation, Methodology, Writing - review and editing; Heike Bähre, Investigation, Methodology, Resources, Writing - review and editing; Roland Seifert, Oliver Scherf-Clavel, Methodology, Resources, Writing - review and editing; Markus Thamm, Conceptualization, Funding acquisition, Project administration, Resources, Supervision, Writing - original draft, Writing - review and editing

### Author ORCIDs

Sinan Kaya-Zeeb http://orcid.org/0000-0001-5349-8787
Markus Thamm http://orcid.org/0000-0003-0480-2206

### Decision letter and Author response

Decision letter https://doi.org/10.7554/eLife.74334.sa1
Author response https://doi.org/10.7554/eLife.74334.sa2

## Additional files

### Supplementary files

• Transparent reporting form

### Data availability

All data generated or analyzed during this study are included in the manuscript and supporting file; Source Data files have been provided for Figures 1,3,4,5,6.

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
