## [Editor Report]

This study is of broad interest to researchers in the field of entomology and physiology. These findings may shed light on at least one mechanism underlying selective advantages conferred to insect species on evolutionary timescales. Though the chemical signal, its source, and recipient tissues underlying thermogenesis are elucidated, hypotheses regarding their downstream effects remain to be substantiated.

---

## [Decision Letter]

**Decision letter after peer review:**

Thank you for submitting your article "Octopamine drives honeybee thermogenesis" for consideration by *eLife*. Your article has been reviewed by 2 peer reviewers, and the evaluation has been overseen by a Reviewing Editor and K VijayRaghavan as the Senior Editor. The reviewers have opted to remain anonymous.

Essential revisions:

Please see the revisions below asked by both reviewers. While the octopamine injection experiment is promising but needs additional experiments outlined in the two reviews to discern details of the process of thermogenesis and support multiple hypotheses in the manuscript.

*Reviewer #1 (Recommendations for the authors):*

The findings are likely to be of real interest to *eLife*.

In order to make this a full manuscript, I recommend that some evidence be provided for the "Hypothetical cascade".

I recommend that the authors investigate flight muscle

i) Fructose 2,6 bis phosphate levels

ii) Pyruvate levels

under the conditions of stimulated and inhibited octopamine signaling employed in this manuscript. This is in order to investigate if active octopamine signaling in flight muscle affects glycolysis in any demonstrable way. Other definitive approaches in this direction would be welcome. Any correlations that may appear in these experiments might support their biochemical hypothesis.

Restatement suggestion:

Line 234: " The data of our study supports the hypothesis that octopamine is necessary and sufficiant for honeybee thermogenesis. "

To make it work out of the context of this paper, this sentence may be reworked. I can see that octopamine signaling is necessary. In the absence of the flight muscles, octopamine would not be sufficient. Too many qualifiers would be needed to apply the phrase "necessary and sufficient" accurately. Therefore I recommend this sentence be rephrased.

In my experience, declarative statements in papers of fair importance in Biology need to be corrected subsequently, in light of new data based in novel more sensitive techniques/environmental conditions etc. I would advise caution against them.

*Reviewer #2 (Recommendations for the authors):*

I consider the investigations of Kaya-Zeeb and colleagues to be very interesting and relevant in principle, as they were able to elucidate an essential aspect of bee life in this way. In most aspects, the manuscript is well written and the experiments are comprehensible. The experiments are mostly adequately and competently performed. Nevertheless, there remain some issues that seem unclear to me or where I see substantial potential for optimization.

1) The authors measure thermogenesis in fixed flies running on a treadmill. It is not clear if this is shivering thermogenesis or not. Can it be distinguished? That is, is thermogenesis permanently activated or not?

2) The conclusion that the OARß2 receptor is primarily responsible for the effects is currently correlative and based on the assumed effectuation (cAMP increase), also the role of cAMP for this activity is not proven. This needs to be worked out.

3) Is there a possibility to analyze the two candidates (receptors) in more detail, e.g. by RNAi experiments with specific receptor knock down allowing more precise identification of the receptors involved in this process.

---

## [Author Response]

Reviewer #1 (Recommendations for the authors):The findings are likely to be of real interest to eLife.In order to make this a full manuscript, I recommend that some evidence be provided for the "Hypothetical cascade".I recommend that the authors investigate flight musclei) Fructose 2,6 bis phosphate levelsii) pyruvate levelsunder the conditions of stimulated and inhibited octopamine signaling employed in this manuscript. This is in order to investigate if active octopamine signaling in flight muscle affects glycolysis in any demonstrable way. Other definitive approaches in this direction would be welcome. Any correlations that may appear in these experiments might support their biochemical hypothesis.

We agree with the reviewer. Unfortunately, we do not have the technical expertise to quantify fructose-2,6-bis-phosphate. However, we have added several analyses and discuss the results accordingly and updated our scheme (now Figure 7):

1. PKA inhibition results in hypothermia in nurse bees and forager bees (L139-145, Figure 6A, L261-262).

2. We established pyruvate quantification (L377-387) and could show that octopamine exposure increases flight muscle pyruvate concentrations (L146-150, Figure 6B, L268270).

3. We can show that the AmGAPDH (codes for the step 6 enzyme in glycolysis) gene expression is increased by both cold stress and octopamine injection (L151-155, Figure 6C-D, L270-274).

Restatement suggestion:Line 234: " The data of our study supports the hypothesis that octopamine is necessary and sufficiant for honeybee thermogenesis. "To make it work out of the context of this paper, this sentence may be reworked. I can see that octopamine signaling is necessary. In the absence of the flight muscles, octopamine would not be sufficient. Too many qualifiers would be needed to apply the phrase "necessary and sufficient" accurately. Therefore I recommend this sentence be rephrased.In my experience, declarative statements in papers of fair importance in Biology need to be corrected subsequently, in light of new data based in novel more sensitive techniques/environmental conditions etc. I would advise caution against them.

We agree with Reviewer 1 and rephrased our statement (L256-257). Consequently, we changed corresponding passages in the entire manuscript (L19, L200-201).

Reviewer #2 (Recommendations for the authors):I consider the investigations of Kaya-Zeeb and colleagues to be very interesting and relevant in principle, as they were able to elucidate an essential aspect of bee life in this way. In most aspects, the manuscript is well written and the experiments are comprehensible. The experiments are mostly adequately and competently performed. Nevertheless, there remain some issues that seem unclear to me or where I see substantial potential for optimization.1) The authors measure thermogenesis in fixed flies running on a treadmill. It is not clear if this is shivering thermogenesis or not. Can it be distinguished? That is, is thermogenesis permanently activated or not?

This is an important question, and we thank the reviewer for addressing it. There was a debate if bees perform shivering or non-shivering thermogenesis or both. Several studies however show that thermogenesis in bees is always enabled by muscle contractions, even if this is not visible externally (no wing/thorax vibration; Esch et al., 1991, Esch and Goller 1991). We added this important information to our manuscript (L37-40)

Furthermore, thermogenesis is not permanently activated. On an individual level, this is more likely to be observed in waves (see Kronenberg and Heller, 1982, J comp physiol,148(1):65–76, Figure 13). This is also happens (with some variation) also in our experimental setup. To demonstrate this, we add a new Figure supplement to Figure 4 (L705).

2) The conclusion that the OARß2 receptor is primarily responsible for the effects is currently correlative and based on the assumed effectuation (cAMP increase), also the role of cAMP for this activity is not proven. This needs to be worked out.

Since Yohimbine has no effect on thermogenesis, we assume that AmOARα1 has no crucial role in thermogenesis (as statet in L234-246). Expression of AmOARα2 was not detectable in the flight muscle. Yohimbine was also active at this receptor subtype. Consequently, the three β octopamine receptors remain as potential candidates. Here, the reviewer is right, we can not discriminate pharmacologically between these candidates.

However, β2 octopamine receptor is the most likely candidate due to its strong expression within the muscle tissues. We have toned down our statement and tried to make it clearer (L204, L234-241, L256-257).

Additionally, we performed an analysis, where we inhibit PKA. Our data clearly show, if cAMP cannot develop its effect (due to occupied PKA binding sides), thermogenesis is disturbed (L139-145, Figure 6A, L261-262). This indirectly supports additionally our hypothesis, that at least one β octopamine receptor subtype is involved in the observed effects, because this receptor subclass is coupled to Gs proteins and by this to PKA.

3) Is there a possibility to analyze the two candidates (receptors) in more detail, e.g. by RNAi experiments with specific receptor knock down allowing more precise identification of the receptors involved in this process.

Experiments in this direction would be great analyses for the future. Unfortunately, we do not have the ability to perform such an analysis currently. Furthermore, we do not know if the corresponding receptor protein expression can be affected especially by RNAi (since we do not have specific antibodies to control for this). On the other hand, even if RNAi supports our hypothesis, we do not know whether the knockdown was specific to the tissue of interest (flight muscle) because RNAi acts systemically and thus affects the whole organism.

Thus we decided to target the receptor proteins directly with pharmacological agents (see Figure 5 and the corresponding discussion L217-246). As mentioned above, our results suggest strongly that AmOARβ2 is the most likely candidate, although we cannot rule out the other β octopamine receptors by 100%.